# Does cranberry extract reduce antibiotic use for symptoms of acute uncomplicated urinary tract infections (CUTI)? A feasibility randomised trial

Oghenekome Gbinigie [1], Julie Allen,[1] Nicola Williams,[1] Michael Moore [2], Alastair D Hay,[3] Carl Heneghan [1], Anne-Marie Boylan [1], Christopher C Butler [1]

¹Nuffield Department of Primary Care Health Sciences, University of Oxford, Oxford, UK
²Primary Care Medical Group, University of Southampton Medical School, Southampton, UK
³Centre for Academic Primary Care, University of Bristol, Bristol, UK

**Correspondence to**
Dr Oghenekome Gbinigie;
oghenekome.gbinigie@phc.ox. ac.uk

## ABSTRACT

**Objectives** To determine the feasibility of conducting a randomised trial of the effectiveness of cranberry extract in reducing antibiotic use by women with symptoms of acute, uncomplicated urinary tract infection (UTI).

**Design** Open-label feasibility randomised parallel group trial.

**Setting** Four general practices in Oxfordshire.

**Participants** Women aged 18 years and above presenting to general practice with symptoms of acute, uncomplicated UTI.

**Interventions** Women were randomly assigned using Research Electronic Data Capture in a 1:1:1 ratio to: (1) immediate antibiotics alone (n=15); (2) immediate antibiotics and immediate cranberry capsules for up to 7 days (n=15); or (3) immediate cranberry capsules and delayed antibiotics for self-initiation in case of non-improvement or worsening of symptoms (n=16).

**Primary and secondary outcome measures** The primary outcome measures were: rate of recruitment of participants; numbers lost to follow-up; proportion of electronic diaries completed by participants; and acceptability of the intervention and study procedures to participants and recruiters. Secondary outcomes included an exploration of differences in symptom burden and antibiotic use between groups.

**Results** Four general practitioner practices (100%) were opened and recruited participants between 1 July and 2 December 2019, with nine study participants recruited per month on average. 68.7% (46/67) of eligible participants were randomised (target 45) with a mean age of 48.4 years (SD 19.9, range 18–81). 89.1% (41/46) of diaries contained some participant entered data and 69.6% (32/46) were fully complete. Three participants (6.5%) were lost to follow-up and two (4.4%) withdrew. Of women randomly assigned to take antibiotics alone (controls), one-third of respondents reported consuming cranberry products (33.3%, 4/12). There were no serious adverse events.

**Conclusions** It appears feasible to conduct a randomised trial of the use of cranberry extract in the treatment of acute, uncomplicated UTI in general practice.

**Trial registration number** ISRCTN Registry (ID: 10399299).

### Strengths and limitations of this study

► This feasibility trial addresses a novel use of cranberry extract in the treatment of *acute*, rather than prevention of *recurrent*, urinary tract infection.
► A pragmatic, open-label randomised trial design was employed.
► In keeping with pragmatic trials, no placebo control was used.

## INTRODUCTION

In light of rising levels of antimicrobial resistance, there has been growing interest in the use of non-antibiotic treatments for common bacterial infections, such as urinary tract infections (UTIs). UTIs are one of the most common bacterial infections seen in primary care,[1] and are nearly always treated with antibiotics.[2] Trials have evaluated non-antibiotic treatments for acute uncomplicated UTIs in primary care,[3–7] and use of ibuprofen has enabled some women to manage UTIs without taking antibiotics, although with a greater symptom burden compared with antibiotic treatment, and more cases of pyelonephritis.[3 6 7]

Cranberry has been extensively evaluated as a preventative agent for UTIs, with mixed results. Up to 27% of women report using cranberry to help treat the symptoms of an acute UTI.[8] Despite this, there is a lack of trials assessing whether cranberry can be used to treat the symptoms of *acute* UTIs, alone, or in combination with antibiotics. There are some mechanistic data to support this approach; urine collected within 6 hours of consumption of cranberry powder containing standardised amounts of proanthocyanidins (PAC) has been shown to exert an anti-adhesion effect against *Escherichia coli* in vitro, in a dose-dependent fashion.[9] PAC

with type A linkages, or the breakdown products after digestion in the gut, is believed to be the active ingredient in cranberry.[10]

The primary objective of this feasibility trial was therefore to determine the feasibility of conducting a randomised clinical trial to determine the effectiveness of cranberry extract in reducing antibiotic use by women with symptoms of acute, uncomplicated UTI in general practice.

## METHODS

### Trial design and participants

This was an open-label, pragmatic, feasibility trial with randomisation in a 1:1:1 ratio between treatment groups in UK primary care. The eligibility criteria have been described in detail elsewhere.[11] In brief, women aged 18 years and above with symptoms of acute, uncomplicated UTI, to whom their healthcare practitioner would normally give a prescription for immediate antibiotics, and presenting to one of four participating general practitioner (GP) practices in Oxfordshire, were eligible. Prospective participants had to be prepared to accept a delayed antibiotic prescription; that is, an antibiotic prescription with advice not to take the antibiotics unless their symptoms did not improve, or worsened. Women were excluded if they had signs of upper or complicated UTI, if they had received antibiotics in the preceding 7 days, if they had a symptom duration of 7 days or more, or if they regularly consumed cranberry products on 5 or more days per week.

Participants were emailed a link to an electronic symptom diary and were asked to complete it daily for 2 weeks. They were asked to rate a number of symptoms on a scale of 0–50, and to record any UTI treatments (study interventions or otherwise) that they were taking. The list of symptoms in the daily diary has been published with the feasibility trial protocol.[11] The scale used was an adaptation of a validated 7-point Likert scale developed by Watson *et al*[12]:

▶ 0=normal/not affected.
▶ 1–9=very little problem.
▶ 10–19=slight problem.
▶ 20–29=moderately bad.
▶ 30–39=bad.
▶ 40–49=very bad.
▶ 50=as bad as it could be.

We used this wider scale as part of the feasibility testing to see whether participants liked having more rating options. We also wondered whether the wider scale might enable better detection of within-participant and between-participant rating variation. A follow-up phone call took place at 2 weeks to encourage diary completion and obtain a minimum data set from participants whose diaries were incomplete. Women were invited for interview at a later date if they had provided consent at the point of trial recruitment to receive information about the interview study. A notes review took place at 4 weeks

to record any clinical contacts that were believed to be related to the original UTI episode, as well as the urine culture result.

Adverse event data were collected through symptom diary entries, through participant reports at the 2-week follow-up call and through notes reviews.

### Interventions

Women in group 1 (control) received a prescription for first-line immediate antibiotics alone. In keeping with pragmatic trial design, the antibiotic prescribed and the duration was at the discretion of the recruiting clinician. In the UK, the usual recommended first-line treatment for acute uncomplicated UTI is a 3-day course of nitrofurantoin.[13] Women assigned to group 2 received an immediate prescription for first-line antibiotics and a course of cranberry capsules to be taken alongside antibiotics until symptom resolution, or up to a maximum of 7 days. The cranberry capsules (Redicran) were supplied free to the study by Indena SpA. This company had no other involvement with the trial. Each capsule contained 18 mg of PAC. Women were advised to take two capsules two times per day at 12-hour intervals, making a total of 72 mg of PAC daily. Ex-vivo studies suggest that consumption of 72 mg PAC has anti-adhesion activity against uropathogenic *E. coli*.[9] Women in group 3 were advised to take immediate cranberry capsules (Redicran) for up to 7 days. They also received a delayed antibiotic prescription to be used if their symptoms did not improve within 3–5 days, or if their symptoms worsened.

### Outcomes

The primary outcomes were feasibility measures,[11] namely, the rate of recruitment of participants, the number of participants lost to follow-up, and the acceptability of trial processes and the quality of data capture using electronic symptom diaries. These data can contribute to sample size calculations for an adequately powered trial, and additionally increase the chance of such a trial being successful. The study was not powered to detect differences between groups. Nevertheless, secondary outcomes included an exploration of between-group differences in measures of symptom burden, antibiotic consumption and adverse events.[11]

Although not made explicit in the protocol, a comparison of the time to feeling fully recovered between groups was made, as prespecified in the statistical analysis plan. These data were already being collected through the electronic diary, which had received ethical approval. A decision to conduct an exploratory analysis of the proportion of participants experiencing 'double sickening' (feeling fully recovered and then subsequently feeling unwell within 2 weeks of trial enrolment) was made after the statistical analysis plan had been completed, but before the trial data had been reviewed or analyses commenced. The concept of double sickening has been noted in acute respiratory illness[14]; we felt that an exploratory analysis of this concept in the context of acute UTI would be novel

and useful. A substantial amendment was approved by the ethics committee after trial recruitment had started to allow the trial team to interview enrolling clinicians to the trial, as part of the study's feasibility assessment.

## Sample size and randomisation

A sample size of 45 participants was chosen to enable estimation of a loss-to-follow-up rate of 20% with a CI of 9.6% to 34.6%.[11] Sample sizes of between 24[15] and 50[16] have been suggested for pilot studies.

The randomisation sequence was generated using Stata V.15.1 (StataCorp, College Station, Texas, USA) by a statistician at the Nuffield Department of Primary Care Health Sciences, University of Oxford, using computer-generated block randomisation with variable block size. The sequence was password protected and accessible only by the statistician. Participants were randomised by recruiting clinicians through the click of a button on the electronic trial registration page, which was created using Research Electronic Data Capture.

## Statistical methods

Unless otherwise stated, primary outcomes are presented as proportions and percentages for each trial group and for the entire trial population. Participants in group 2 and group 3 were compared separately with participants in group 1 (the control group).

The number of antibiotic courses (or part thereof) consumed by participants in each group at 2 weeks and 4 weeks was compared using Poisson regression and reported using incidence rate ratios. The difference in the duration of symptoms rated moderately bad or worse between groups was compared using negative binomial regression; incidence rate ratios are reported and a Kaplan-Meier graph plotted (online supplemental file 1). The mean symptom burden scores (from day 2 to day 4 inclusive) were compared between groups using linear regression, and reported using mean differences. The difference in time to feeling fully recovered between groups was compared using the Cox proportional-hazards model; HRs are presented and a Kaplan-Meier graph was plotted (online supplemental file 1). Each estimate was adjusted for study site.

A risk ratio and 95% CI was used to estimate the difference in the proportion of participants consuming a course of antibiotics (or part thereof) between groups at 2 weeks and 4 weeks, the difference in the proportion of adverse events between groups at both 2 weeks and 4 weeks, and to compare the proportion of participants in each group experiencing double sickening. It was not possible to use log binomial regression, as originally planned, to compare the proportion of participants consuming antibiotics between groups as the model would not converge. A risk ratio and 95% CI was used in lieu of Fisher's exact test (as originally planned) to estimate the difference in proportion of adverse events between groups in order to generate 95% CIs rather than p values. As this study was not powered to detect differences between groups, we had prespecified that p values would not be reported.

Intention-to-treat analysis was conducted for all analyses and estimates are presented with their 95% CIs. Due to the small sample size and the fact that this is a feasibility study, no subgroup or sensitivity analyses were conducted.

## Patient and public involvement

Four patient and public involvement (PPI) contributors have been closely involved with the study from the outset, two of whom sit on the Trial Steering Committee. During trial conception and development, they confirmed that the topic chosen was important. They critiqued all public-facing trial documentation to ensure that it was free of jargon, and their feedback was incorporated. They were also involved in the design of the electronic diary, helping to ensure that it was user-friendly. PPI contributors unanimously felt that the diary would not be onerous for participants; and suggested it may make the experience of participating in the trial more meaningful, stating "… completing the symptom diary hooks people in. It makes them feel that they are contributing." The results of the study have been shared with PPI contributors, allowing them to give their views on ways to explain the findings, which have been incorporated into the discussion. PPI contributors are also helping to inform the dissemination of the study findings.

# RESULTS

## Baseline data

Baseline information was completed electronically by participants upon accessing the electronic diary on day 1 of enrolment, prior to inputting their day 1 symptom ratings. The mean age of participants was 48.3 years (range: 18–81 years). The mean age of women in the immediate cranberry and delayed antibiotics group was lower than the other two groups. Most women had experienced a previous UTI (85.4%, 35/41). A total of 75.6% (31/41) of respondents were expecting antibiotics. Sixty-one per cent (25/41) believed that cranberry could help their symptoms, while 24.4% (10/41) believed that their symptoms could improve without antibiotics (table 1).

## Primary outcomes

Forty-six women were recruited to the trial, surpassing the target of 45 (see figure 1). Of the 21 women who were eligible but declined to take part, the most common reason for declining was not wishing to be randomised to receive delayed antibiotics (42.9%, 9/21). The first participant was recruited on 11 July 2019 and the final participant was recruited on 2 December 2019, with the final follow-up taking place on 30 December 2019. The findings of qualitative interviews with 14 trial participants and 8 trial recruiters will be presented elsewhere.

A total of 89.1% (41/46) of electronic diaries had some data entered by participants, while 69.6% (32/46) of

**Table 1** Baseline characteristics of participants

| Proportion (%), unless otherwise specified | Group 1 (n=15) | Group 2 (n=15) | Group 3 (n=16) | Total (n=46) |
|---|---|---|---|---|
| Age in years, mean (SD) (range) | 52.4 (21.4) (18–81) | 52.4 (22.4) (19–81) | 40.8 (14.0) (20–63) | 48.3 (19.9) (18–81) |
| Previous UTI | | | | |
| Yes | 12/15 (80%) | 12/13 (92.3%) | 11/13 (84.6%) | 35/41 (85.4%) |
| No | 3/15 (20%) | 1/13 (7.7%) | 2/13 (15.4%) | 6/41 (14.6) |
| Missing | 0/15 | 2/15 | 3/16 | 5/46 |
| Previous UTI in the past year | | | | |
| Yes | 7/15 (46.7%) | 6/13 (46.2%) | 6/13 (46.2%) | 19/41 (46.3%) |
| No | 5/15 (33.3%) | 6/13 (46.2%) | 5/13 (38.5%) | 16/41 (39.0%) |
| N/A (have never had a previous UTI) | 3/15 (20%) | 1/13 (7.7%) | 2/13 (15.4%) | 6/41 (14.6%) |
| Missing | 0/15 | 2/15 | 3/16 | 5/46 |
| If had a UTI in the past year, number of months since last UTI—mean (SD), median (range) | 5.4 (4.5), 4 (0–12) n=7 | 5 (4.2), 5 (0–11) n=5 | 7 (3.6), 7 (3–12) n=5 | 5.8 (4.2), 5 (0–12) n=17 |
| Can't remember | 0/15 (0%) | 1/13 (7.7%) | 1/12 (8.3%) | 2/40 (5%) |
| Not applicable | 8/15 (53.3%) | 7/13 (53.9%) | 6/12 (50%) | 21/40 (52.5%) |
| Missing | 0/15 | 2/15 | 4/16 | 6/46 |
| Number of UTIs in the past year | | | | |
| 0 | 5/15 (33.3%) | 7/13 (53.9%) | 4/13 (30.8%) | 16/41 (39.0%) |
| 1 | 3/15 (20%) | 2/13 (15.4%) | 5/13 (38.5%) | 10/41 (24.4%) |
| 2 | 2/15 (13.3%) | 2/13 (15.4%) | 1/13 (7.7%) | 5/41 (12.2%) |
| 3 or more | 2/15 (13.3%) | 1/13 (7.7%) | 1/13 (7.7%) | 4/41 (9.8%) |
| Not applicable (have never had a previous UTI) | 3/15 (20%) | 1/13 (7.7%) | 2/13 (15.4%) | 6/41 (14.6%) |
| Missing | 0/15 | 2/15 | 3/16 | 5/46 |
| Were you expecting antibiotic treatment? | | | | |
| Yes | 13/15 (86.7%) | 10/13 (76.9%) | 8/13 (61.5%) | 31/41 (75.6%) |
| No | 0/15 (0%) | 1/13 (7.7%) | 1/13 (7.7%) | 2/41 (4.9%) |
| Unsure | 2/15 (13.3%) | 2/13 (15.4%) | 4/13 (30.8%) | 8/41 (19.5%) |
| Missing | 0/15 | 2/15 | 3/16 | 5/46 |
| Were you expecting tests/investigations? | | | | |
| Yes | 12/15 (80%) | 10/13 (76.9%) | 8/13 (61.5%) | 30/41 (73.2%) |
| No | 3/15 (20%) | 3/13 (23.1%) | 4/13 (30.8%) | 10/41 (24.4%) |

Continued

**Table 1** Continued

| Proportion (%), unless otherwise specified | Group 1 (n=15) | Group 2 (n=15) | Group 3 (n=16) | Total (n=46) |
|---|---|---|---|---|
| Unsure | 0/15 (0%) | 0/13 (0%) | 1/13 (7.7%) | 1/41 (2.4%) |
| Missing | 0/15 | 2/15 | 3/16 | 5/46 |
| **Were you expecting advice?** | | | | |
| Yes | 12/15 (80%) | 11/12 (91.7%) | 13/13 (100%) | 36/40 (90%) |
| No | 2/15 (13.3%) | 1/12 (8.3%) | 0/13 (0%) | 3/40 (7.5%) |
| Unsure | 1/15 (6.7%) | 0/12 (0%) | 0/13 (0%) | 1/40 (2.5%) |
| Missing | 0/15 | 3/15 | 3/16 | 6/46 |
| **Were you expecting something else?** | | | | |
| Yes | 0/15 (0%) | 1/13 (7.7%)—information and prescription (1 participant) | 0/13 (0%) | 1/41 (2.4%) |
| No | 15/15 (100%) | 10/13 (76.9%) | 11/13 (84.6%) | 36/41 (87.8%) |
| Unsure | 0/15 (0%) | 2/13 (15.4%) | 2/13 (15.4%) | 4/41 (9.8%) |
| Missing | 0/15 | 2/15 | 3/16 | 5/46 |
| **Was your last UTI treated with antibiotics?** | | | | |
| Yes | 8/15 (53.3%) | 9/13 (69.2%) | 10/13 (76.9%) | 27/41 (65.9%) |
| No | 4/15 (26.7%) | 3/13 (23.1%) | 1/13 (7.7%) | 8/41 (19.5%) |
| Not applicable | 3/15 (20%) | 1/13 (7.7%) | 2/13 (15.4%) | 6/41 (14.6%) |
| Missing | 0/15 | 2/15 | 3/16 | 5/46 |
| **Was your last UTI treated with something other than antibiotics?** | | | | |
| Yes | 2/15 (13.3%)—cystitis sachets (2 participants) | 1/13 (7.7%)—cystitis sachets (1 participant) | 0/13 (0%) | 3/41 (7.3%) |
| No | 10/15 (66.7%) | 11/13 (84.6%) | 11/13 (84.6%) | 32/41 (78.1%) |
| Not applicable | 3/15 (20%) | 1/13 (7.7%) | 2/13 (15.4%) | 6/41 (14.6%) |
| Missing | 0/15 | 2/15 | 3/16 | 5/46 |
| **Did you receive no treatment for your last UTI?** | | | | |
| Yes | 1/15 (6.7%) | 0/13 (0%) | 0/13 (0%) | 1/41 (2.4%) |
| No | 11/15 (73.3%) | 12/13 (92.3%) | 11/13 (84.6%) | 34/41 (82.9%) |
| Not applicable | 3/15 (20%) | 1/13 (7.7%) | 2/13 (15.4%) | 6/41 (14.6%) |
| Missing | 0/15 | 2/15 | 3/16 | 5/46 |
| **Unable to remember how last UTI was treated** | | | | |

Continued

**Table 1**  Continued

| Proportion (%), unless otherwise specified | Group 1 (n=15) | Group 2 (n=15) | Group 3 (n=16) | Total (n=46) |
|---|---|---|---|---|
| Yes | 1/15 (6.7%) | 1/13 (7.7%) | 1/13 (7.7%) | 3/41 (7.3%) |
| No | 11/15 (73.3%) | 11/13 (84.6%) | 10/13 (76.9%) | 32/41 (78.1%) |
| Not applicable | 3/15 (20%) | 1/13 (7.7%) | 2/13 (15.4%) | 6/41 (14.6%) |
| Missing | 0/15 | 2/15 | 3/16 | 5/46 |
| Treatments tried before consulting primary care | | | | |
| *Cranberry products* | | | | |
| Yes | 2/15 (13.3%) | 3/13 (23.1%) | 3/13 (23.1%) | 8/41 (19.5%) |
| No | 13/15 (86.7%) | 10/13 (76.9%) | 10/13 (76.9%) | 33/41 (80.5%) |
| Missing | 0/15 | 2/15 | 3/16 | 5/46 |
| *Other fruit juice* | | | | |
| Yes | 1/15 (93.3%) | 1/13 (7.7%) | 0/13 (0%) | 2/41 (4.9%) |
| No | 14/15 (93.3%) | 12/13 (92.3%) | 13/13 (100%) | 39/41 (95.1%) |
| Missing | 0/15 | 2/15 | 3/16 | 5/46 |
| *Bicarbonate solution* | | | | |
| Yes | 0/15 (0%) | 0/13 (0%) | 0/13 (0%) | 0/41 (0%) |
| No | 15/15 (100%) | 13/13 (100%) | 13/13 (100%) | 41/41 (100%) |
| Missing | 0/15 | 2/15 | 3/16 | 5/46 |
| *Sodium/potassium citrate (cystitis sachets)* | | | | |
| Yes | 3/15 (20%) | 7/13 (53.9%) | 1/13 (7.7%) | 11/41 (26.8%) |
| No | 12/15 (80%) | 6/13 (46.2%) | 12/13 (92.3%) | 30/41 (73.2%) |
| Missing | 0/15 | 2/15 | 3/16 | 5/46 |
| *Uvacin/uva ursi (bearberry)* | | | | |
| Yes | 0/15 (0%) | 0/13 (0%) | 0/13 (0%) | 0/41 (0%) |
| No | 15/15 (100%) | 13/13 (100%) | 13/13 (100%) | 41/41 (100%) |
| Missing | 0/15 | 2/15 | 3/16 | 5/46 |
| *Pain relief (eg, paracetamol/ibuprofen)* | | | | |
| Yes | 6/15 (40%) | 7/13 (53.9%) | 4/13 (30.8%) | 17/41 (41.5%) |
| No | 9/15 (60%) | 6/13 (46.2%) | 9/13 (69.2%) | 24/41 (58.5%) |
| Missing | 0/15 | 2/15 | 3/16 | 5/46 |
| *Other treatment used?* | | | | |

**Table 1** Continued

| Proportion (%), unless otherwise specified | Group 1 (n=15) | Group 2 (n=15) | Group 3 (n=16) | Total (n=46) |
|---|---|---|---|---|
| Yes | 2/15 (13.3%)—water (2 participants) | 2/13 (15.4%)—tea tree in bath (1 participant); sachets (1 participant) | 2/13 (15.4%)—water (2 participants) | 6/41 (14.6%) |
| No | 13/15 (86.7%) | 11/13 (84.6%) | 11/13 (84.6%) | 35/41 (85.4%) |
| Missing | 0/15 | 2/15 | 3/16 | 5/46 |
| *None* | | | | |
| Yes | 7/15 (46.7%) | 2/13 (15.4%) | 5/13 (38.5%) | 14/41 (34.2%) |
| No | 8/15 (53.3%) | 11/13 (84.6%) | 8/13 (61.5%) | 27/41 (65.9%) |
| Missing | 0/15 | 2/15 | 3/16 | 5/46 |
| Number of days symptomatic | | | | |
| 0 | 1/15 (6.7%) | 2/13 (15.4%) | 1/13 (7.7%) | 4/41 (9.8%) |
| 1 | 3/15 (20%) | 1/13 (7.7%) | 2/13 (15.4%) | 6/41 (14.6%) |
| 2 | 4/15 (26.7%) | 3/13 (23.1%) | 5/13 (38.5%) | 12/41 (29.3%) |
| 3 | 2/15 (13.3%) | 2/13 (15.4%) | 4/13 (30.8%) | 8/41 (19.5%) |
| >3 | 5/15 (33.3%) | 5/13 (38.5%) | 1/13 (7.7%) | 11/41 (26.8%) |
| Missing | 0/15 | 2/15 | 3/16 | 5/46 |
| Belief that cranberry can help symptoms | | | | |
| Yes | 10/15 (66.7%) | 7/13 (53.9%) | 8/13 (61.5%) | 25/41 (61%) |
| No | 1/15 (6.7%) | 0/13 (0%) | 0/13 (0%) | 1/41 (2.4%) |
| Unsure | 4/15 (26.7%) | 6/13 (46.2%) | 5/13 (38.5%) | 15/41 (36.6%) |
| Missing | 0/15 | 2/15 | 3/16 | 5/46 |
| Belief that symptoms can get better without antibiotics | | | | |
| Yes | 5/15 (33.3%) | 1/13 (7.7%) | 4/13 (30.8%) | 10/41 (24.4%) |
| No | 4/15 (26.7%) | 4/13 (30.8%) | 2/13 (15.4%) | 10/41 (24.4%) |
| Unsure | 6/15 (40%) | 8/13 (61.5%) | 7/13 (53.9%) | 21/41 (51.2%) |
| Missing | 0/15 | 2/15 | 3/16 | 5/46 |

Group 1—immediate antibiotics alone (control); group 2—immediate antibiotics and immediate cranberry; group 3—immediate cranberry and delayed antibiotics.
N/A, not applicable; UTI, urinary tract infection.

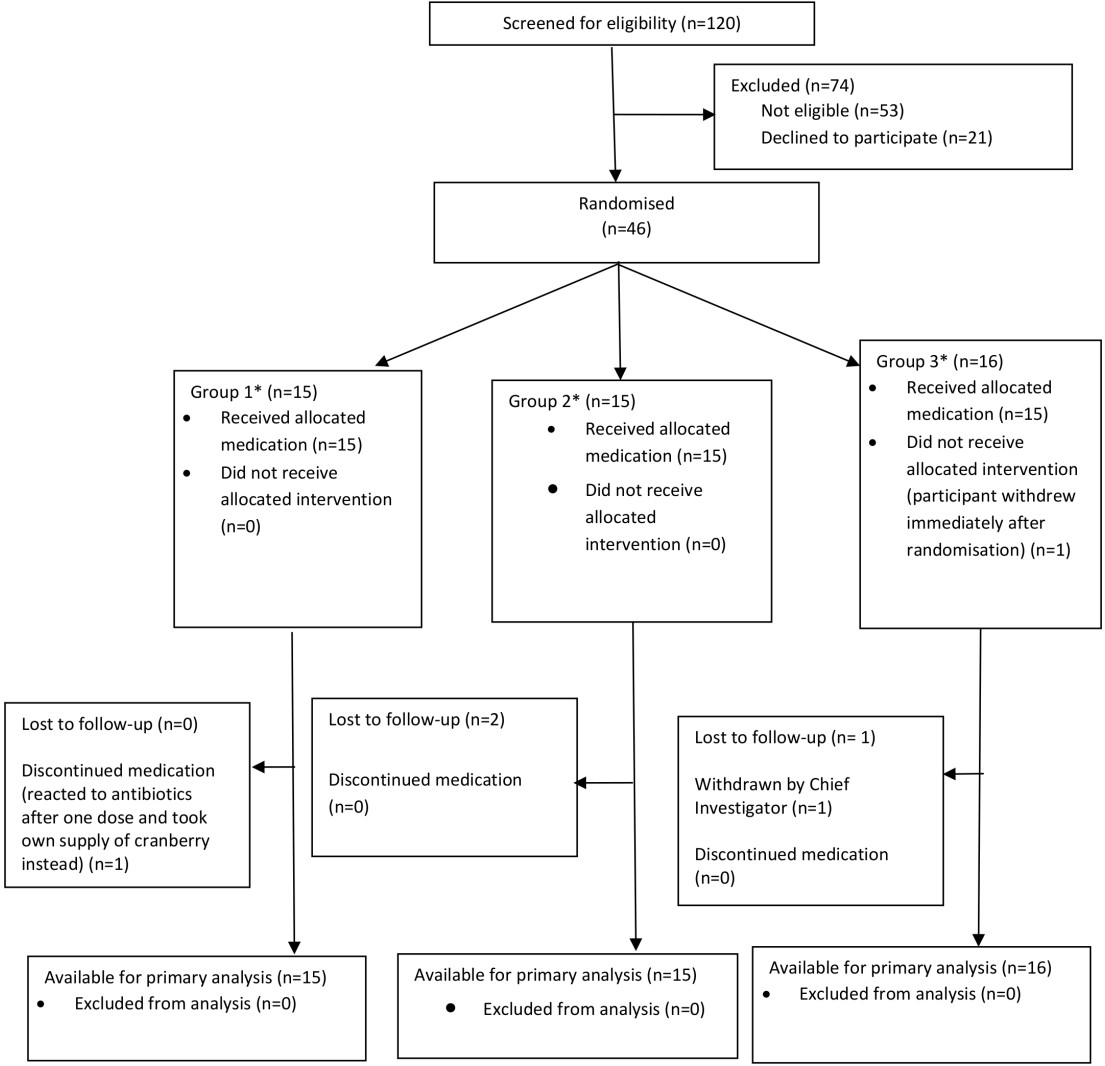

**Figure 1** CONSORT diagram showing the flow of trial participants. *Group 1—immediate antibiotics alone (control); group 2—immediate antibiotics and immediate cranberry capsules; group 3—immediate cranberry capsules and delayed antibiotics. CONSORT, Consolidated Standards of Reporting Trials.

electronic diaries were fully complete (table 2). There were no electronic diary failures necessitating the issue of paper diaries. A mean of 9.2 participants were recruited per month, which exceeded predicted recruitment of between four and eight participants each month. All GP practices who were invited to recruit to the trial agreed to participate (100%, 5/5), however, one GP practice withdrew prior to site opening as they predicted that they would no longer have capacity to support the trial.

Only one participant reported a side effect or problem related to consuming cranberry capsules in the electronic diary (table 2). Few participants reported experiencing problems completing the electronic diary (8.6%, 3/35) or rating their symptoms using the scale of 0–50 (11.4%, 4/35).

Three participants were lost to follow-up (figure 1). Loss to follow-up was determined to have occurred when a participant did not make any electronic diary entries and it was not possible to contact them at the 2-week follow-up call.

Two participants withdrew from the study. One participant was consented and randomly assigned to immediate cranberry capsules with delayed antibiotics. As soon as she received this assignment, she decided to withdraw from the study and did not receive the cranberry capsules. Another participant was consented and also randomised to immediate cranberry and delayed antibiotics. The day after being enrolled, she emailed the trial team asking to be removed from the mailing list (links to the electronic diary were sent by email). In light of this, the chief investigator withdrew the participant from the study.

Participants were able to provide free-text comments on the final day (day 14) of diary completion. Free-text boxes allowed women to provide general comments about the trial, as well as comments specifically about difficulties using the electronic diary, difficulties using the rating scale and any problems related to consuming cranberry capsules. Fourteen women provided comments.

Some women provided general feedback, including being 'happy to help' with the study and finding the diary

**Table 2** Primary outcomes

| Proportion (%), unless otherwise specified | Group 1 (n=15) | Group 2 (n=15) | Group 3 (n=16) | Total (n=46) |
|---|---|---|---|---|
| Number of diaries with some data points entered by participant by end of participant follow-up | 15 (100%) | 13 (86.7%) | 13 (81.3%) | 41 (89.1%) |
| Number of diaries fully completed by participant by end of participant follow-up | 13 (86.7%) | 10 (66.7%) | 9 (56.3%) | 32 (69.6%) |
| Paper diaries issued due to failure of electronic diary | 0 (0%) | 0 (0%) | 0 (0%) | 0 (0%) |
| Number randomised as a proportion of the target recruitment number | 46/45 (102.2%) | | | |
| Number randomised as a proportion of invited participants | 46/120 (38.3%) | | | |
| Number randomised as a proportion of eligible participants | 46/67 (68.7%) | | | |
| No of patients invited | | | | 120 |
| No of patients eligible | | | | 67 |
| No of participants enrolled | 15 | 15 | 16 | 46 |
| No of participants randomised | 15 | 15 | 16 | 46 |
| No of recruited per month (n) | | | | 9.2 |
| Number of participants lost to follow-up | 0 (0%) | 2 (13.3%) | 1 (6.3%) | 3 (6.5%) |
| Number of participants withdrawing from study | 0 (0%) | 0 (0%) | 2 (12.5%) | 2 (4.4%) |
| Side effects or problems related to cranberry consumption | | | | |
| Yes | 0/13 (0%) | 0/10 (0%) | 1/11 (9.1%) | 1/34 (2.9%) |
| No | 1/13 (7.7%) | 10/10 (100%) | 10/11 (90.9%) | 21/34 (61.8%) |
| Not applicable | 12/13 (92.3%) | 0/10 (0%) | 0/11 (0%) | 12/34 (35.3%) |
| Missing | 2/15 | 5/15 | 5/16 | 12/46 |
| Problems completing the electronic diary | | | | |
| No problem | 14/14 (100%) | 9/10 (90%) | 9/11 (81.8%) | 32/35 (91.4%) |
| Some problems | 0/14 (0%) | 1/10 (10%) | 2/11 (18.2%) | 3/35 (8.6%) |
| Lots of problems | 0/14 (0%) | 0/10 (0%) | 0/11 (0%) | 0/35 (0%) |
| Missing | 1/15 | 5/15 | 5/16 | 11/46 |
| Problems using the scale of 0–50 | | | | |
| No problem | 12/14 (85.7%) | 9/9 (100%) | 10/12 (83.3%) | 31/35 (88.6%) |
| Some problems | 2/14 (14.3%) | 0/9 (0%) | 2/12 (16.7%) | 4/35 (11.4%) |
| Lots of problems | 0/14 (0%) | 0/9 (0%) | 0/12 (0%) | 0/35 (0%) |
| Missing | 1/15 | 6/15 | 4/16 | 11/46 |
| Proportion of invited GP practices agreeing to participate in the trial | 5/5 (100%) | | | |

Continued

**Table 2** Continued

| Proportion (%), unless otherwise specified | Group 1 (n=15) | Group 2 (n=15) | Group 3 (n=16) | Total (n=46) |
|---|---|---|---|---|
| Proportion of invited GP practices opened and recruiting to the trial | 4/5 (80%) | | | |
| Acceptability of the study to participants and recruiters, participants' experience of having a UTI and thoughts on self-help for UTIs | Interviews with trial participants and recruiters will be reported elsewhere. | | | |

Group 1—immediate antibiotics alone (control); group 2—immediate antibiotics and immediate cranberry; group 3—immediate cranberry and delayed antibiotics.
GP, general practitioner; UTI, urinary tract infection.

'very easy to complete'. Two participants described some difficulty keeping up with filling in the diary 'Just time and keeping up with it that is all. Happy to have taken part.' *(41-8, Immediate cranberry and delayed antibiotics).* Another participant mentioned that she did not receive a link to complete the final set of diary questions, and one woman was unclear about the guidance regarding the need to continue completing the diary once symptoms had resolved for 2 consecutive days. Two women commented that the scale of 0–50 was 'too wide a range'.

One woman stated that she was sceptical about cranberry at first, but found that it 'helped a lot' (32-11, Immediate cranberry and delayed antibiotics). One participant found that the combination of antibiotic with cranberry cleared her infection 'quite quickly' (32-14, Immediate antibiotics and immediate cranberry), while another found that the antibiotic/cranberry combination resulted in her taking a shorter course of antibiotics than usual—'Nice to only take three days of antibiotics not a week' (23-3, Immediate antibiotics and immediate cranberry). One woman stated that despite starting with cranberry, she had to take antibiotics on day 2 as she was in pain (41-2, Immediate cranberry and delayed antibiotics).

Three adverse events were picked up from free-text comments (see the Harms section below).

### Secondary outcomes

Approximately one-third of respondents reported contacting a healthcare practitioner within 2 weeks of being enrolled (36.4%, 12/33). Most of these women contacted their GP practice (GP or nurse at their surgery)—one woman reported liaising with a pharmacist, and another contacted an out-of-hours GP. No participants reported requiring Accident and Emergency or secondary care input. Just over half (54.3%, 25/46) of the urine samples at the time of enrolment were positive (organism growth greater than $10^5$/CFU (colony-forming unit)).

Exploratory analyses of effectiveness were conducted, while recognising that the study was not powered for these analyses (table 3). For this reason, p values have not been reported and any differences detected between groups are inconclusive.

There was a 20% reduction in the risk of consuming a course of antibiotics (or part thereof) in the immediate cranberry group (group 3), compared with the immediate antibiotics alone group (group 1—controls). This was the case at both 2 weeks and 4 weeks (risk ratio 0.8; 95% CI: 0.6 to 1.0). The number of courses of antibiotics consumed at 2 weeks and at 4 weeks was reduced in groups 2 (immediate antibiotics and immediate cranberry) and 3 (immediate cranberry and delayed antibiotics), compared with the control group (adjusted incidence rate ratios: 0.7; 95% CI: 0.4 to 1.4 and 0.6; 95% CI: 0.3 to 1.2 for group 2 and group 3, respectively). Participants in groups 2 and 3 had a longer duration of symptoms rated moderately bad or worse, compared with those in the control group (adjusted incidence rate ratios: 1.2; 95% CI: 0.7 to 2.1 and 2.9; 95% CI 1.7 to 5.1 for group 2 and group 3, respectively). The mean symptom burden (from day 2 to day 4 inclusive) was reduced in group 2 participants (adjusted mean difference: 0.6; 95% CI: –4.0 to 5.3) and increased in group 3 participants (adjusted mean difference: 7.9; 95% CI 2.6 to 13.2), compared with the control group. Similarly, the time to feeling recovered was reduced in group 2 (adjusted HR 1.7; 95% CI: 0.7 to 4.1) but increased in group 3 (adjusted HR 0.6; 95% CI: 0.2 to 1.4), compared with controls. Of women randomly assigned to take antibiotics alone (controls), one-third of respondents reported consuming cranberry products (33.3%, 4/12) (table 4).

An exploratory analysis was conducted to compare the proportion of participants experiencing 'double sickening' between groups at 2 weeks. Double sickening was defined as a participant selecting the relevant checkbox in the electronic diary to confirm that they felt fully recovered, then subsequently indicating through checkbox responses that they no longer felt fully recovered. Given the short period of time over which this was assessed (2 weeks from enrolment), it is likely that these double sickening episodes represent incompletely treated infection, rather than a new infection. Compared with the control group, group 2 participants had a reduced risk of experiencing double sickening (risk ratio 0.6; 95% CI: 0.2 to 2.0), while none of the group 3 participants experienced double sickening.

**Table 3** Exploratory effectiveness outcomes

| | Proportion (%) of participants consuming a course of antibiotics (or part thereof) at 2 weeks (risk ratio (95% CI))* | Proportion (%) of participants consuming a course of antibiotics (or part thereof) at 4 weeks (risk ratio (95% CI))* | Mean (SD) number of courses of antibiotics consumed at 2 weeks (adjusted incidence rate ratio (95% CI))*† | Mean (SD) number of courses of antibiotics consumed at 4 weeks (adjusted incidence rate ratio (95% CI))*† | Median number of days of symptoms rated moderately bad or worse (adjusted incidence rate ratio (95% CI))*† | Mean (SD) symptom burden from day 2 to 4 inclusive (adjusted mean difference (95% CI))*†‡ | Median (IQR) days to feeling fully recovered (adjusted HR (95% CI))*† | Proportion (%) of participants experiencing double sickening up to 2 weeks (risk ratio (95% CI))* |
|---|---|---|---|---|---|---|---|---|
| Group 1 (control) n=15 | 15/15 (100) Missing: 0/15 | 15/15 (100) Missing: 0/15 | 1.5 (0.5) Missing: 0/15 | 1.6 (0.6) Missing: 0/15 | 2 (1–2) Missing: 0/15 | 6.3 (3.4) Missing: 1/15 | 5 (4–6) Missing: 0/15 | 6/14 (42.9) Missing: 1/15 |
| Group 2 n=15 | 13/13 (100) (1 (1 to 1)) Missing: 2/15 | 13/13 (100) (1 (1 to 1)) Missing: 2/15 | 1.1 (0.3) (0.7 (0.4 to 1.4)) Missing: 2/15 | 1.2 (0.4) (0.7 (0.4 to 1.4)) Missing: 2/15 | 2 (1–3) (1.2 (0.7 to 2.1)) Missing: 2/15 | 6.2 (8.0) (0.6 (−4.0 to 5.3)) Missing: 4/15 | 4 (3–6) (1.7 (0.7 to 4.1)) Missing: 2/15 | 3/11 (27.3) (0.6 (0.2 to 2.0)) Missing: 4/15 |
| Group 3 n=16 | 11/14 (78.6) (0.8 (0.6 to 1.0)) Missing: 2/16 | 11/14 (78.6) (0.8 (0.6 to 1.0)) Missing: 2/16 | 0.9 (0.6) (0.6 (0.3 to 1.2)) Missing: 2/16 | 0.9 (0.6) (0.6 (0.3 to 1.2)) Missing: 2/16 | 4 (2–8) (2.9 (1.7 to 5.1)) Missing: 4/16 | 11.2 (8.5) (7.9 (2.6 to 13.2)) Missing: 7/16 | 13 (5–9) (0.6 (0.2 to 1.4)) Missing: 3/16 | 0/12 (0) (0 (N/A)) Missing: 4/16 |

Group 1—immediate antibiotics alone (control); group 2—immediate antibiotics and immediate cranberry; group 3—immediate cranberry and delayed antibiotics.

*Group 2 and group 3 were compared with group 1 (controls).

†Model adjusted for site.

‡The mean symptom burden for each participant was calculated by summing the total ratings for all symptoms between day 2 and day 4 (inclusive) and dividing this by the number of symptom ratings made by the participant between day 2 and day 4 (inclusive).

N/A, not applicable.

**Table 4** Secondary outcomes

| Proportion (%), unless otherwise specified | Group 1 (n=15) | Group 2 (n=15) | Group 3 (n=16) | Total (n=46) |
|---|---|---|---|---|
| Any of the following treatments used in the past 2 weeks (not including study medication) | | | | |
| *Cranberry products* | | | | |
| Yes | 4/12 (33.3%) | 2/9 (22.2%) | 2/9 (22.2%) | 8/30 (26.7%) |
| No | 8/12 (66.7%) | 7/9 (77.8%) | 7/9 (77.8%) | 22/30 (73.3%) |
| Missing | 3/15 | 6/15 | 7/16 | 16/46 |
| *Other fruit juice* | | | | |
| Yes | 0/12 (0%) | 0/9 (0%) | 0/9 (0%) | 0/30 (0%) |
| No | 12/12 (100%) | 9/9 (100%) | 9/9 (100%) | 30/30 (100%) |
| Missing | 3/15 | 6/15 | 7/16 | 16/46 |
| *Bicarbonate solution* | | | | |
| Yes | 0/12 (0%) | 0/9 (0%) | 0/9 (0%) | 0/30 (0%) |
| No | 12/12 (100%) | 9/9 (100%) | 9/9 (100%) | 30/30 (100%) |
| Missing | 3/15 | 6/15 | 7/16 | 16/46 |
| *Sodium/potassium citrate (cystitis sachets)* | | | | |
| Yes | 0/12 (0%) | 0/9 (0%) | 1/9 (11.1%) | 1/30 (3.3%) |
| No | 12/12 (100%) | 9/9 (100%) | 8/9 (88.9%) | 29/30 (96.7%) |
| Missing | 3/15 | 6/15 | 7/16 | 16/46 |
| *Uvacin/uva ursi (bearberry)* | | | | |
| Yes | 0/12 (0%) | 0/9 (0%) | 0/9 (0%) | 0/30 (0%) |
| No | 12/12 (100%) | 9/9 (100%) | 9/9 (100%) | 30/30 (100%) |
| Missing | 3/15 | 6/15 | 7/16 | 16/46 |
| *Pain relief (eg, paracetamol/ibuprofen)* | | | | |
| Yes | 5/12 (33.3%) | 1/9 (11.1%) | 5/9 (55.6%) | 11/30 (36.7%) |
| No | 7/12 (58.3%) | 8/9 (88.9%) | 4/9 (44.4%) | 19/30 (63.3%) |
| Missing | 3/15 | 6/15 | 7/16 | 16/46 |
| *Other* | | | | |
| Yes | 0/12 (0%) | 0/9 (0%) | 0/9 (0%) | 0/30 (0%) |
| No | 12/12 (100%) | 9/9 (100%) | 9/9 (100%) | 30/30 (100%) |
| Missing | 3/15 | 6/15 | 7/16 | 16/46 |
| *None* | | | | |
| Yes | 5/12 (41.7%) | 6/9 (66.7%) | 3/9 (33.3%) | 14/30 (46.7%) |

**Table 4** Continued

| Proportion (%), unless otherwise specified | Group 1 (n=15) | Group 2 (n=15) | Group 3 (n=16) | Total (n=46) |
|---|---|---|---|---|
| No | 7/12 (58.3%) | 3/9 (33.3%) | 6/9 (66.7%) | 16/30 (53.3%) |
| Missing | 3/15 | 6/15 | 7/16 | 16/46 |
| Time off paid work or normal activities | | | | |
| Yes | 5/13 (38.5%) | 2/10 (20%) | 3/9 (33.3%) | 10/32 (31.3%) |
| No of days, median (IQR) | 1 (1–2) | 5 (2–8) | 2 (1–2) | 2 (1–2) |
| No | 8/13 (61.5%) | 8/10 (80%) | 6/9 (66.7%) | 22/32 (68.8%) |
| Missing | 2/15 | 5/15 | 7/16 | 14/46 |
| Healthcare professional consulted at GP practice | | | | |
| Yes | 6/13 (46.2%) | 1/8 (12.5%) | 5/12 (41.7%) | 12/33 (36.4%) |
| No | 7/13 (53.9%) | 7/8 (87.5%) | 7/12 (58.3%) | 21/33 (63.6%) |
| Missing | 2/15 | 7/15 | 4/16 | 13/46 |
| *GP at surgery* | | | | |
| Yes | 3/13 (23.1%) | 0/8 (0%) | 2/12 (16.7%) | 5/33 (15.2%) |
| No of times, median (IQR) | 1 (1–1) | N/A | 1 (1–1) | 1 (1–1) |
| No | 3/13 (23.1%) | 1/8 (12.5%) | 3/12 (25%) | 7/33 (21.2%) |
| Not applicable | 7/13 (53.9%) | 7/8 (87.5%) | 7/12 (58.3%) | 21/33 (63.6%) |
| Missing | 2/15 | 7/15 | 4/16 | 13/46 |
| *Nurse at surgery* | | | | |
| Yes | 2/13 (15.4%) | 1/8 (12.5%) | 1/12 (8.3%) | 4/33 (12.1%) |
| No of times, median (IQR) | 1 (1–1) | 2 (2–2) | 1 (1–1) | 1 (1–1.5) |
| No | 4/13 (30.8%) | 0/8 (0%) | 4/12 (33.3%) | 8/33 (24.2%) |
| Not applicable | 7/13 (53.9%) | 7/8 (87.5%) | 7/12 (58.3%) | 21/33 (63.6%) |
| Missing | 2/15 | 7/15 | 4/16 | 13/46 |
| *GP at home* | | | | |
| Yes | 0/13 (0%) | 0/8 (0%) | 0/12 (0%) | 0/33 (0%) |
| No of times, median (IQR) | N/A | N/A | N/A | N/A |
| No | 6/13 (46.2%) | 1/8 (12.5%) | 5/12 (41.7%) | 12/33 (36.4%) |
| Not applicable | 7/13 (53.9%) | 7/8 (87.5%) | 7/12 (58.3%) | 21/33 (63.6%) |
| Missing | 2/15 | 7/15 | 4/16 | 13/46 |
| *OOH doctor* | | | | |

| **Table 4** Continued | | | | |
| --- | --- | --- | --- | --- |
| **Proportion (%), unless otherwise specified** | **Group 1 (n=15)** | **Group 2 (n=15)** | **Group 3 (n=16)** | **Total (n=46)** |
| Yes | 1/13 (7.7%) | 0/8 (0%) | 0/12 (0%) | 1/33 (3.0%) |
| No of times, median (IQR) | 1 (1–1) | N/A | N/A | 1 (1–1) |
| No | 5/13 (38.5%) | 1/8 (12.5%) | 5/12 (41.7%) | 11/33 (33.3%) |
| Not applicable | 7/13 (53.9%) | 7/18 (87.5%) | 7/12 (58.3%) | 21/33 (63.6%) |
| Missing | 2/15 | 7/15 | 4/16 | 13/46 |
| *Other* | | | | |
| Yes | 0/13 (0%) | 0/8 (0%) | 2/12 (16.7%)— pharmacist (1 participant), phoned GP (1 participant) | 2/33 (6.1%) |
| No of times, median (IQR) | N/A | N/A | 1 (1–1) | 1 (1–1) |
| No | 6/13 (46.2%) | 1/8 (12.5%) | 3/12 (25%) | 10/33 (30.3%) |
| Not applicable | 7/13 (53.9%) | 7/8 (87.5%) | 7/12 (58.3%) | 21/33 (63.6%) |
| Missing | 2/15 | 7/15 | 4/16 | 13/46 |
| Healthcare professional consulted at A&E | | | | |
| Yes | 0/13 (0%) | 0/10 (0%) | 0/12 (0%) | 0/35 (0%) |
| No | 13/13 (100%) | 10/10 (100%) | 12/12 (100%) | 35/35 (100%) |
| Missing | 2/15 | 5/15 | 4/16 | 11/46 |
| No of times, median (IQR) | N/A | N/A | N/A | N/A |
| Seen by a specialist (excluding hospital admission) | | | | |
| Yes | 0/13 (0%) | 0/10 (0%) | 0/12 (0%) | 0/35 (0%) |
| No | 13/13 (100%) | 10/10 (100%) | 12/12 (100%) | 35/35 (100%) |
| Missing | 2/15 | 5/15 | 4/16 | 11/46 |
| No of times, median (IQR) | N/A | N/A | N/A | N/A |
| Admitted to hospital | | | | |
| Yes | 0/13 (0%) | 0/10 (0%) | 0/12 (0%) | 0/35 (0%) |
| No | 13/13 (100%) | 10/10 (100%) | 12/12 (100%) | 35/35 (100%) |
| Missing | 2/15 | 5/15 | 4/16 | 11/46 |
| No of nights, median (IQR) | N/A | N/A | N/A | N/A |
| Positive midstream urine culture | | | | |
| Yes | 10/15 (66.7%) | 7/15 (46.7%) | 8/16 (50%) | 25/46 (54.3%) |
| No | 5/15 (33.3%) | 8/15 (53.3%) | 8/16 (50%) | 21/46 (45.7%) |

Continued

**Table 4** Continued

| Proportion (%), unless otherwise specified | Group 1 (n=15) | Group 2 (n=15) | Group 3 (n=16) | Total (n=46) |
|---|---|---|---|---|
| Missing | 0/15 | 0/15 | 0/16 | 0/46 |
| Proportion with a positive midstream urine culture | 54.3% | | | |

All outcomes reported in relation to the 2 weeks following trial enrolment.
Group 1—immediate antibiotics alone (control); group 2—immediate antibiotics and immediate cranberry; group 3—immediate cranberry and delayed antibiotics.
A&E, Accident and Emergency; GP, general practitioner; N/A, not applicable; OOH, out-of-hours.

**Table 5** List of adverse events

| Adverse event number | Group | Adverse event | Outcome | Severity | Relationship to study product | Comments |
|---|---|---|---|---|---|---|
| 1 | Immediate antibiotics alone | Tummy upset that started 3 days after starting ciprofloxacin (second course of antibiotics) | Resolved | Mild | Not related | Not applicable |
| 2 | Immediate antibiotics alone | Reaction after taking one dose of antibiotic | Resolved | Uncertain | Not related | Not applicable |
| 3 | Immediate antibiotics and immediate cranberry | Suprapubic pain, dysuria, backache, cloudy urine and single vomit | Resolved | Moderate | Possibly related | Possible pyelonephritis from description of symptoms, although not reported as such by healthcare practitioner. MSU positive— *Escherichia coli* $10^5$ CFU/mL. |
| 4 | Immediate antibiotics and immediate cranberry | Vomiting after second course of nitrofurantoin started | Resolved | Mild | Possibly related | Vomit after second course of antibiotics started. Antibiotics stopped and participant encouraged to push fluids. Symptoms settled without further antibiotics being prescribed. |
| 5 | Immediate cranberry and delayed antibiotics | Feeling unwell, ache in lower back | Resolved | Moderate | Possibly related | Symptoms unlikely to be related to the study product, as the participant was recruited in the morning and the symptoms started the same day in the afternoon. Participant reported feeling fully recovered on day 2 and did not contact healthcare practitioner up to 28 days. MSU positive >*E.coli* $10^5$ CFU/mL. |
| 6 | Immediate cranberry and delayed antibiotics | Felt sick | Resolved | Mild | Possibly related | Participant mentioned that she was unsure whether the sickness symptom was related to cranberry or her UTI. |
| 7 | Immediate cranberry and delayed antibiotics | General feeling of being unwell, terrible backache and stomachache at times, headache and nausea | Resolved | Moderate | Possibly related | Possible pyelonephritis, however, no growth on MSU, reported feeling fully recovered on day 5, and the participant did not contact a healthcare practitioner up to 28 days. |

CFU, colony-forming unit; MSU, midstream urine; UTI, urinary tract infection.

## Harms

Seven adverse events occurred (table 5). Adverse events 3 and 4 occurred in the same participant. Two adverse event descriptions sounded like *possible* pyelonephritis (numbers 3 and 7). However, one of these adverse events occurred in a participant whose midstream urine was negative, who reported feeling fully recovered on day 5 of the study and who did not contact a healthcare practitioner up to 28 days after enrolment. The second occurred in a participant who subsequently contacted her GP practice, however, her clinical syndrome was not reported as pyelonephritis by the consulting healthcare practitioner and the participant's symptoms settled with increased fluid intake alone. There were no serious adverse events.

The difference in the proportion of adverse events between groups was compared, although the feasibility trial was not powered to detect differences between groups. At 2 weeks, the point estimates indicated an increased risk of experiencing an adverse event for participants in group 2 (risk ratio 1.2; 95% CI: 0.1 to 16.7) and group 3 (risk ratio 3.5; 95% CI: 0.4 to 29.4) compared with the control group, although the 95% CIs were wide. At 4 weeks, the risk of experiencing an adverse event in group 2 was lower than for the control group (risk ratio 0.6; 95% CI: 0.1 to 5.7). The risk remained higher for participants in group 3 compared with controls at 4 weeks (risk ratio 1.7; 95% CI: 0.3 to 8.8) (online supplemental file 2).

## DISCUSSION

In this open-label, feasibility randomised trial, we recruited beyond target and at a rate faster than predicted. Overall, participants found the trial procedures and interventions acceptable; few reported problems consuming cranberry capsules or using the electronic symptom diary.

### Strengths and weaknesses

To our knowledge, this is the first trial to use cranberry capsules alone and combined with antibiotics as an *acute* treatment for UTI. The trial processes were designed, in discussion with PPI contributors, to be minimally burdensome for participants and recruiters, which may have contributed to the faster than anticipated recruitment, and to recruiting beyond target.

Using a participant-completed diary led to missing data, and in particular, incomplete baseline data. We also collected very limited demographic data. Missing data means that we do not have a full picture of all of the participants' experiences and outcomes. However, insights gained from the trial combined with the qualitative interviews (to be reported elsewhere) should help to optimise future trial design and minimise missing data in a future study.

Some may describe the open-label design as a weakness. However, this was designed to be a pragmatic trial with procedures closely aligned with real-world practice. We were therefore not prescriptive about the antibiotic

issued, nor the duration, recognising that clinicians' prescribing is influenced by a host of factors, including a patient's medical history and UTI history. Despite this, we appreciate that there is a lack of standardisation of antibiotic prescribing between groups. Furthermore, participants who missed diary entries were able to complete the diary retrospectively; retrospective entries are likely to be less accurate than real-time diary completion.

We did not control participants' consumption of other sources of PAC with type A linkages, namely blueberry. However, unless participants regularly consumed large amounts of blueberry, it is unlikely that this would have had a significant impact on the results. Interestingly, despite advice not to, one-third of women in the immediate antibiotics alone group reported consuming cranberry products during the trial. This highlights the challenge of trying to control external influences in a trial, particularly when this is in conflict with a participant's usual behaviour.

### Comparison with the existing literature

No study participants developed pyelonephritis. A number of trials of non-antibiotic treatments have reported higher levels of pyelonephritis in participants not assigned to an antibiotic arm.[3 5–7] The absence of cases of pyelonephritis in this trial may well have been due to the small sample size, but may also have resulted from all participants receiving a prescription for antibiotics, whether immediate or delayed antibiotics. In a trial of 382 women with uncomplicated UTIs evaluating the use of ibuprofen and the herbal extract uva ursi, all participants received delayed antibiotics and there were no cases of pyelonephritis.[4]

The proportion of participants lost to follow-up or withdrawing from this feasibility trial (11%) is roughly comparable with analogous trials, which report between 1.3%[17] and 16%.[7] Of note, loss to follow-up is defined differently in different studies. The proportion of participants with a positive urine culture (54.3%) was lower than the proportion reported in some analogous trials (range of 32%[4]–80%[17]). Of note, this trial used a cut-off threshold for urine culture positivity of $10^5$ CFU/mL, while most other trials were aligned with European guidance and used a threshold of greater than $10^2$ or $10^3$ CFU/mL.

While not powered to detect differences between groups, in line with the findings of similar trials, participants in this feasibility trial not receiving immediate antibiotics (group 3) consumed fewer antibiotics but had a higher symptom burden. However, participants in group 3 were less likely to experience double sickening. Interestingly, Gagyor *et al*[6] found that significantly fewer participants receiving ibuprofen compared with antibiotics experienced UTI recurrence between days 15 and 28 (percentage mean difference −5.3% (95% CI: −10.2% to −0.4%), p=0.049). However, the difference between groups did not reach statistical significance when all recurrent UTIs from day 1 to day 28 were assessed

(percentage mean difference −2.8% (95% CI: −8.7% to 3.1%), p=0.41).

A Cochrane review in 2012 found that overall, treatment with cranberry did not significantly reduce *recurrent* UTIs (n=13 studies; risk ratio 0.86; 95% CI: 0.71 to 1.04).[18] This was also the case in a subgroup analysis that focused on women with UTI (n=4 studies; risk ratio 0.74; 95% CI: 0.42 to 1.31). However, the review authors acknowledged that many included studies had high rates of participant withdrawal/loss to follow-up of up to 55%, largely attributed to difficulty adhering to drinking large volumes of cranberry juice on a daily basis. Furthermore, there was a lack of standardisation and poor reporting of the amount of cranberry extract—and specifically of PAC—in the interventions used. In this feasibility trial, we used cranberry capsules with a standardised amount of PAC, for short-term use in the context of acute UTI. These key differences may have contributed to better adherence of participants to the study intervention.

### Implications for clinicians, researchers and policymakers

This feasibility trial was not powered to detect differences between groups, thus, no definite inferences about the effectiveness or safety of cranberry in the context of acute uncomplicated UTI can be made. If cranberry extract is found to be safe and effective in the management of acute UTIs through robust and adequately powered trials, it could represent an important health intervention for this common condition and has the potential to significantly reduce antibiotic consumption.

There is a need for adequately powered, randomised clinical trials to provide definitive evidence of whether cranberry extract is effective in reducing antibiotic use and managing symptoms of acute UTI. While this feasibility trial does not provide evidence of the effectiveness or safety of cranberry extract in the context of acute UTI, the exploratory analyses suggest that it would be worthwhile to explore these as yet unanswered questions through an adequately powered trial. Such trials should use cranberry products with standardised amounts of PAC. Participants should be willing to stop all non-trial cranberry products for the duration of the study; as a minimum, the use of non-trial cranberry products by participants should be captured.

### CONCLUSION

It is feasible to conduct a randomised trial of similar design to this study to assess whether cranberry extract can help reduce antibiotic use for symptoms of acute, uncomplicated UTI. The exploratory, underpowered analyses suggest that cranberry (alone and combined with antibiotics) *may* have a beneficial impact on certain measures of antibiotic usage and symptoms. These findings support the conduct of further, definitive research in this area.

**Acknowledgements** The authors would like to thank all study participants, participating GP practices and staff in Oxford University's Primary Care Clinical Trials Unit who were responsible for electronic database development (Ms Rui Zhao and Mr Luis Castello). The authors would also like to thank members of the Trial Steering Committee (TSC): Dr Akke Vellinga (TSC Chair and epidemiologist); Ms Julia Hamer-Hunt (PPI representative); Dr Bernard Gudgin (PPI representative); Dr Merlin Willcox (academic clinical lecturer and GP); Dr Emma Thomas-Jones (research fellow and senior trials manager) and Dr Kathy Hughes (clinical lecturer and GP). We thank Dr Sarah Tonkin-Crine for helpful feedback on the manuscript, and PPI contributors for their involvement with the trial.

**Contributors** OG was responsible for the research questions, writing the protocol, the design and running of the feasibility trial, data analysis and writing the manuscript. OG is the guarantor of the manuscript. JA is a senior trial manager who assisted with running the feasibility trial and contributed to the manuscript. NW is a senior statistician who contributed to the protocol and statistical analysis plan, and verified statistical analyses of primary outcomes. CCB, MM, ADH and CH helped to develop the research questions and study design and contributed to the manuscript. A-MB was a member of the TMG and contributed to the manuscript.

**Funding** This work was supported by the NIHR School for Primary Care Research (grant reference number: SPCR-2014-10043). Indena SpA supplied Redicran capsules to the trial at no cost. OG's time is funded by the Wellcome Trust (grant reference number: 203921). CH received funding support from the NIHR SPCR Evidence Synthesis Working Group (project 390) and the NIHR Oxford BRC. A-MB receives funding from the NIHR School for Primary Care Research.

**Disclaimer** The funders have no role in study design, manuscript submission, or collection, management, analysis or interpretation of study data. The views are those of the authors and not necessarily those of the NIHR or Department of Health and Social Care, nor the Wellcome Trust. For the purpose of Open Access, the author has applied a CC BY public copyright licence to any Author Accepted Manuscript version arising from this submission.

**Competing interests** OG receives funding from the National Institute of Health Research (NIHR) SPCR and the Wellcome Trust. CCB is a senior investigator of the NIHR; clinical director of the University of Oxford Primary Care and Vaccines Collaborative Clinical Trials Unit; clinical director of the NIHR Oxford Community Medtech and Invitro diagnostics Cooperative; and salaried general practitioner for the Cwm Taf Morgannwg University Health Board. He has received funding from many public funding bodies for primary care research related to the management of common infections. He received payment for contributing to Advisory Boards for Pfizer in 2019, Roche Diagnostics in 2020, and for contributing to an Advisory Board for Janssen Pharmaceuticals about respiratory syncytial virus treatment and vaccination from Janssen Pharmaceuticals in 2020, and holds an unrestricted grant from Janssen Pharmaceuticals for contributing to research on respiratory syncytial virus. CH reports he has received expenses and fees for his media work. He is director of the CEBM at the University of Oxford, and editor-in-chief of BMJ Evidence-Based Medicine and an NIHR senior investigator. JA, NW, MM, ADH and A-MB have no conflicts of interest to declare.

**Patient consent for publication** Not required.

**Ethics approval** Informed consent was obtained from all participants prior to entry into the study. In accordance with the Declaration of Helsinki, the CUTI Study was approved by the South Central Oxford B Ethics Committee (REC reference: 18/SC/0673) and the Health Research Authority (IRAS Project ID: 249672).

**Provenance and peer review** Not commissioned; externally peer reviewed.

**Data availability statement** Data are available upon reasonable request. Relevant anonymised data are available upon reasonable request made to the corresponding author.

## ORCID iDs

Oghenekome Gbinigie http://orcid.org/0000-0002-2963-4491
Michael Moore http://orcid.org/0000-0002-5127-4509
Carl Heneghan http://orcid.org/0000-0002-1009-1992
Anne-Marie Boylan http://orcid.org/0000-0001-8187-0742
Christopher C Butler http://orcid.org/0000-0002-0102-3453

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
