## [Reviewer comments · BMJ Open]

ARTICLE DETAILS

TITLE (PROVISIONAL)	Does cranberry extract reduce antibiotic use for symptoms of acute uncomplicated urinary tract infections (UTI)? A feasibility randomised trial.
AUTHORS	Gbinigie, Oghenekome; Allen, Julie; Williams, Nicola; Moore, Michael; Hay, Alastair; Heneghan, Carl; Boylan, Anne-Marie; Butler, Christopher C.

VERSION 1 – REVIEW

REVIEWER	Nikesh Thiruchelvam Department of Urology Addenbrookes Hospital Cambridge University Hospitals NHS Trust
REVIEW RETURNED	24-Nov-2020

GENERAL COMMENTS	Well written and clear feasibility. Some minor revision would be helpful to clarify the below points: Why use a likert scale of 0-50 rather than 0-10 Why was there a significant age difference in group 3 (specifically a pre-menopausal group who may have a different UTI risk and resolution profile to the post-menopausal women of group 1 and 2) and why was their diary completion so poor? The pragmatic nature of the trial and the reasoning behind open - label design is very clear in the text.
--

REVIEWER	Kevin C. Maki Midwest Biomedical Research, Addison, IL, USA I have conducted clinical trials and published in the area of cranberry and UTI sponsored by Ocean Spray Cranberries.
REVIEW RETURNED	04-Dec-2020

GENERAL COMMENTS	This paper describes the results of a feasibility randomized trial to evaluate the use of a cranberry extract in the treatment of acute, uncomplicated UTI in general practice. The authors also report results for secondary outcomes to explore potential differences in symptoms and antibiotic use between treatment groups that included immediate antibiotics alone, immediate antibiotics and immediate cranberry capsules (taken for up to 7 days), or immediate cranberry capsules and delayed antibiotics (to be self-initiated if symptoms did not improve or worsened). The authors concluded that it is feasible to conduct such a trial. There are a few points that I think need to be clarified in the manuscript, but, overall, I think the paper is excellent. Please see suggested revisions below.
---

Specific Comments to the Authors

1. Introduction: although the authors discuss cranberry as a potential treatment for UTI, suggest that they also refer to it as a potential preventive agent for UTI.
2. Methods: has the list of symptoms in the daily diary been published elsewhere? If yes, suggest providing a reference citation.
3. Methods: it is not entirely clear how long the participants were instructed to complete the electronic daily symptom diaries, and whether this was also used to record their intake of the prescribed antibiotics and cranberry capsules, etc. The methods indicate that a follow-up call took place at 2 weeks to encourage diary completion, and the Results (Lines 274-275) state "participants were able to provide free-text comments on the final day (day 14) of diary completion..." but there are follow-up data reported for both 2 and 4 weeks, suggesting data were collected beyond 2 weeks. Please clarify. Also, suggest including in the Methods a description of adverse event collection.
4. Methods, Lines 155-156: suggest that, "This data was" should be written as "These data were." Line 163: suggest rather than "recruiters" that they could be described as "enrolling clinicians."
5. Methods, Lines 204-215: suggest defining PPI (it appears to represent Patient and Public Involvement); also is this paragraph necessary, and, if yes, is this the appropriate placement of it, or would this be better placed as part of the Acknowledgements section?
6. Table 1: perhaps consider re-wording the following questions in the follow-up randomized clinical trial as they might be confusing to the participants: "Did you receive no treatment for your last UTI?" and "Unable to remember how last UTI was treated?" Perhaps consider the alternative wording of: "Did you receive treatment for your last UTI?" and "Do you remember how your last UTI was treated?"
7. In Table 2, the number of participants withdrawing from the study in Group 3 is reported to be 2, and the 2 withdrawals are also described in Lines 266-272. However, in Figure 1, there is only an explanation of 1 patient who withdrew immediately after randomization. The other participant described in the text, who withdrew the day after being enrolled, needs to be described in Figure 1 also.
8. Line 293: suggest replacing "one lady" with either "one woman" or "one participant."
9. Lines 300-304: were these healthcare practitioner contacts that occurred within 2 weeks of being enrolled related to a UTI, or were these for any reason? Lines 304-305: suggest specifying the time that the urine samples were collected, i.e., at the time of enrollment.
10. Table 3: suggest adding the notation to the table that Group 1 was the referent for the risk ratios.
11. Lines 344-352: because double sickening is reported in Table 3, suggest moving up this paragraph that describes double sickening to the area where the rest of Table 3 is described, i.e., around Line 336.
12. Line 357: please verify if MSU (midstream urine) has been defined earlier in the paper.
13. Table 5: suggest that the Adverse Events be re-organized by treatment group in the table, i.e., show Group 1-Immediate antibiotics alone 1st, followed by Group 2-Immediate antibiotics

	and immediate cranberry, and finally Group 3-Immediate cranberry and delayed antibiotics. (Please note that the text would also need to be revised since the Adverse Event “Number” would change.) Also, suggest providing details in the Comments section for the “Reaction” the patient experienced after taking one dose of antibiotic. Also, there are some details for the 2 patients with possible pyelonephritis in the text that could be added to the table, i.e., that the patient reported feeling fully recovered on day 5 of the study and that the participant’s symptoms settled with increased fluid intake alone.
--	--

REVIEWER	Holly Fisher Newcastle University, UK
REVIEW RETURNED	18-Dec-2020

GENERAL COMMENTS	I enjoyed reading this write up of a feasibility randomised trial that considers using cranberry extract in the management of acute uncomplicated UTI. The paper is very well written and adheres closely to the recommendations set out in the CONSORT extension to randomised pilot and feasibility trials (Eldridge et al. 2016). The authors have clearly demonstrated that it would be feasible to conduct a larger, definitive RCT in this area. My comments below are minor in nature and should not take long to address. I would have expected the feasibility specific CONSORT checklist to have been completed rather than the general one as per the BMJ Open guidance (although most of the items are overlapping and there are very few omissions in the paper). Comments on the abstract (aligned with the CONSORT 2010 checklist of information to include when reporting a pilot or feasibility randomized trial in a journal or conference abstract):  • Add “parallel group” to the design section of the abstract • Add the method by which patients were randomised to groups • Add the number of participants analysed in each group for the pilot objectives • I think it would be clearer to say, “Four GP practices (100%) were opened to recruitment between...” given it was patients who were recruited to the trial rather than GP practices • Please add a sentence on harms. Comments on the main text (aligned with the CONSORT 2010 checklist of information to include when reporting a pilot or feasibility trial):  • Outcomes section: The authors clearly describe the changes made to the planned outcomes and analysis. However, no justification is given for why an analysis of double sickening was included at such a late stage after the SAP was finalised? (CONSORT item 6B requires justification). • Statistical methods: I could not find the Kaplan Meier graphs the authors refer to in either the main text or supplementary material. • Statistical methods: Although made very clear in the results and tables that comparisons involve comparing the two intervention groups to the control group separately, stating this explicitly in the statistical methods would aid the reader.
--

VERSION 1 – AUTHOR RESPONSE

Reviewers' comments	Authors' response
Reviewer 1 - Comment 1 Well written and clear feasibility. Some minor revision would be helpful to clarify the below points: Why use a likert scale of 0-50 rather than 0-10.	We used a 0-50 scale (an adaptation of a validated 7 point Likert scale) as part of the feasibility testing. We wondered whether participants might prefer a wider scale (and consequently more rating options). We also wondered whether the wider scale might allow us to be more sensitive in detecting within- and between-participant rating variation. The following has now been added to the manuscript to explain this (please see lines 127-129): “We used this wider scale as part of the feasibility testing to see whether participants liked having more rating options. We also wondered whether the wider scale might enable better detection of within- and between-participant rating variation.”
Reviewer 1 - Comment 2 Why was there a significant age difference in group 3 (specifically a pre-menopausal group who may have a different UTI risk and resolution profile to the post-menopausal women of group 1 and 2) and why was their diary completion so poor?	It is unclear why the mean age of participants in group 3 was younger than in the other two groups, and why their diary completion was poorer. There is no indication that randomisation failed. It is therefore likely that the observed differences in group 3 arose due to random chance as a result of the small numbers in the feasibility trial.
Reviewer 1 – Comment 3 The pragmatic nature of the trial and the reasoning behind open -label design is very clear in the text.	We thank the reviewer for this comment.
Reviewer 2 – Comment 1 This paper describes the results of a feasibility randomized trial to evaluate the use of a cranberry extract in the treatment of acute, uncomplicated UTI in general practice. The authors also report results for secondary outcomes to explore potential differences in symptoms and antibiotic use between treatment groups that included immediate antibiotics alone, immediate antibiotics and immediate cranberry capsules (taken for up to 7 days), or immediate cranberry capsules and delayed antibiotics (to be self-initiated if symptoms did not improve or worsened). The authors concluded that it is feasible to conduct such a trial. There are a few points that I think need to be clarified in the manuscript, but, overall, I think	We thank the reviewer for this positive endorsement of our manuscript.

the paper is excellent. Please see suggested revisions below.	
Reviewer 2 – Comment 2 1. Introduction: although the authors discuss cranberry as a potential treatment for UTI, suggest that they also refer to it as a potential preventive agent for UTI.	In line with the reviewer’s comment, we have amended lines 86 to 87 of the introduction such that it now reads “Cranberry has been extensively evaluated as a preventative agent for UTIs...”
Reviewer 2 – Comment 3 2. Methods: has the list of symptoms in the daily diary been published elsewhere? If yes, suggest providing a reference citation.	The list of symptoms in the daily diary has been published as part of the protocol paper. In line with the reviewer’s comment, a reference citation has been added within the manuscript - See page 5, lines 117 to 118: “The list of symptoms in the daily diary has been published with the feasibility trial protocol (11)”
Reviewer 2 – Comment 4 3. Methods: it is not entirely clear how long the participants were instructed to complete the electronic daily symptom diaries, and whether this was also used to record their intake of the prescribed antibiotics and cranberry capsules, etc. The methods indicate that a follow-up call took place at 2 weeks to encourage diary completion, and the Results (Lines 274-275) state “participants were able to provide free-text comments on the final day (day 14) of diary completion...” but there are follow-up data reported for both 2 and 4 weeks, suggesting data were collected beyond 2 weeks. Please clarify. Also, suggest including in the Methods a description of adverse event collection.	Participants were asked to complete the daily diaries for two weeks. In order to make this clearer in the text, we have amended lines 115 to 116 such that they now read “Participants were emailed a link to an electronic symptom diary and were asked to complete it daily for 2 weeks.” Participants were asked to rate their symptoms and record their intake of any UTI interventions that they were taking, including antibiotics and cranberry capsules (please see lines 116 to 117 of the manuscript): “They were asked to rate a number of symptoms on a scale of 0 to 50, and to record any UTI treatments (study interventions or otherwise) that they were taking.” The 4 week follow-up data was collected through notes reviews. The following has now been added to the manuscript to make this clear (see lines 133 to 134): “A notes review took place at 4 weeks to record any clinical contacts that were believed to be related to the original UTI episode, as well as the urine culture result.” In line with the reviewer’s suggestion, we have added the following to the methods to describe adverse event collection (please see lines 135-136): “Adverse event data were collected through symptom diary entries, through

	participant reports at the 2 week follow-up call, and through notes reviews.”
Reviewer 2 – Comment 5 4. Methods, Lines 155-156: suggest that, “This data was” should be written as “These data were.” Line 163: suggest rather than “recruiters” that they could be described as “enrolling clinicians.”	We have adjusted the text in line with the reviewer’s suggestions – please see lines 164-165 and line 173.
Reviewer 2 – Comment 6 5. Methods, Lines 204-215: suggest defining PPI (it appears to represent Patient and Public Involvement); also is this paragraph necessary, and, if yes, is this the appropriate placement of it, or would this be better placed as part of the Acknowledgements section?	In line with the reviewer’s comment, we have now added ‘(PPI)’ after Patient and Public Involvement (See line 216). We feel that it is important to include a PPI section in the manuscript. The PPI section has been placed in the methods section in line with the BMJ Open submission guidance, but can move it elsewhere if the Editor feels this is appropriate.
Reviewer 2 – Comment 7 6. Table 1: perhaps consider re-wording the following questions in the follow-up randomized clinical trial as they might be confusing to the participants: “Did you receive no treatment for your last UTI?” and “Unable to remember how last UTI was treated?” Perhaps consider the alternative wording of: “Did you receive treatment for your last UTI?” and “Do you remember how your last UTI was treated?”	We very much appreciate these suggestions from the reviewer and will strongly consider incorporating these changes when planning a follow-up randomized clinical trial.
Reviewer 2 – Comment 8 7. In Table 2, the number of participants withdrawing from the study in Group 3 is reported to be 2, and the 2 withdrawals are also described in Lines 266-272. However, in Figure 1, there is only an explanation of 1 patient who withdrew immediately after randomization. The other participant described in the text, who withdrew the day after being enrolled, needs to be described in Figure 1 also.	Figure 1 has been updated in line with this comment. The following has been added: “Withdrawn by Chief Investigator (n=1)”
Reviewer 2 – Comment 9 8. Line 293: suggest replacing “one lady” with either “one woman” or “one participant.”	In line with this comment, we have replaced “one lady” with “one woman.” (please see line 306)
Reviewer 2 – Comment 10	The healthcare practitioner contacts were related to the original UTI episode. We have

Lines 300-304: were these healthcare practitioner contacts that occurred within 2 weeks of being enrolled related to a UTI, or were these for any reason? Lines 304-305: suggest specifying the time that the urine samples were collected, i.e., at the time of enrollment.	now clarified this in the manuscript through the text added at lines 133 to 134 “A notes review took place at 4 weeks to record any clinical contacts that were believed to be related to the original UTI episode...” In line with the reviewer’s comment, “at the time of enrolment” has been added to lines 317 to 318.
Reviewer 2 – Comment 11 10. Table 3: suggest adding the notation to the table that Group 1 was the referent for the risk ratios.	A notation has now been added to Table 3 to make it clear than Group 1 was the referent for the risk ratios.
Reviewer 2 – Comment 12 11. Lines 344-352: because double sickening is reported in Table 3, suggest moving up this paragraph that describes double sickening to the area where the rest of Table 3 is described, i.e., around Line 336.	We have moved the paragraph on double sickening to lines 352 to 360, in line with the reviewer’s suggestion.
Reviewer 2 – Comment 13 12. Line 357: please verify if MSU (midstream urine) has been defined earlier in the paper.	Mid-stream urine has now been written in full before the abbreviation – please see line 381.
Reviewer 2 – Comment 14 13. Table 5: suggest that the Adverse Events be re-organized by treatment group in the table, i.e., show Group 1-Immediate antibiotics alone 1st, followed by Group 2- Immediate antibiotics and immediate cranberry, and finally Group 3-Immediate cranberry and delayed antibiotics. (Please note that the text would also need to be revised since the Adverse Event “Number” would change.) Also, suggest providing details in the Comments section for the “Reaction” the patient experienced after taking one dose of antibiotic. Also, there are some details for the 2 patients with possible pyelonephritis in the text that could be added to the table, i.e., that the patient reported feeling fully recovered on day 5 of the study and that the participant’s symptoms settled with increased fluid intake alone.	The adverse event table has been adjusted such that the adverse events appear in group order. The adverse event numbers within the manuscript text have been adjusted accordingly (See lines 378 and 380). Unfortunately, we do not have further information about the specifics of the ‘reaction’ that the patient experienced after taking one dose of antibiotic. We have now added more detail to the comment section for the possible pyelonephritis case: “reported feeling fully recovered on day 5” has been added to the comment for adverse event number seven.

Reviewer 3 – Comment 1 I enjoyed reading this write up of a feasibility randomised trial that considers using cranberry extract in the management of acute uncomplicated UTI. The paper is very well written and adheres closely to the recommendations set out in the CONSORT extension to randomised pilot and feasibility trials (Eldridge et al. 2016). The authors have clearly demonstrated that it would be feasible to conduct a larger, definitive RCT in this area. My comments below are minor in nature and should not take long to address.	We thank the reviewer for this positive endorsement of our manuscript.
Reviewer 3 – Comment 2 I would have expected the feasibility specific CONSORT checklist to have been completed rather than the general one as per the BMJ Open guidance (although most of the items are overlapping and there are very few omissions in the paper).	In line with the reviewer’s suggestion, the CONSORT extension for feasibility trials has been completed and uploaded.
Reviewer 3 – Comment 3 Comments on the abstract (aligned with the CONSORT 2010 checklist of information to include when reporting a pilot or feasibility randomized trial in a journal or conference abstract):  • Add “parallel group” to the design section of the abstract • Add the method by which patients were randomised to groups  • Add the number of participants analysed in each group for the pilot objectives • I think it would be clearer to say, “Four GP practices (100%) were opened to recruitment between....” given it was patients who were recruited to the trial rather than GP practices • Please add a sentence on harms. 	In line with the reviewer’s advice, ‘parallel group’ has been added to the design section of the abstract (please see line 30). The method of randomisation has been added to the abstract, please see lines 34 to 35: “Women were randomly assigned using Research Electronic Data Capture (REDCap)...” The number of participants analysed in each group for the pilot objectives has now been added to the abstract; please see lines 35 to 38. The first sentences of the abstract results now reads: “Four GP practices (100%) were opened and recruited participants between 1st July and 2nd December 2019...” (please see lines 44 to 45) The following sentence on harms has been added to the abstract (please see line 51): “There were no serious adverse events.”
Reviewer 3 – Comment 4	Further justification has now been added to the manuscript – please see lines 170-171: “...we

Comments on the main text (aligned with the CONSORT 2010 checklist of information to include when reporting a pilot or feasibility trial):  Outcomes section: The authors clearly describe the changes made to the planned outcomes and analysis. However, no justification is given for why an analysis of double sickening was included at such a late stage after the SAP was finalised? (CONSORT item 6B requires justification). 	felt that an exploratory analysis of this concept in the context of acute UTI would be novel and useful.”
Reviewer 3 – Comment 5  Statistical methods: I could not find the Kaplan Meier graphs the authors refer to in either the main text or supplementary material. 	The Kaplan Meier graphs have now been submitted as supplementary material and reference has been made to them within the manuscript (please see lines 195 and 199).
Reviewer 3 – Comment 6 Statistical methods: Although made very clear in the results and tables that comparisons involve comparing the two intervention groups to the control group separately, stating this explicitly in the statistical methods would aid the reader.	An explanatory sentence has now been added to the statistical methods, in line with the reviewer’s suggestion (see lines 189-190): “Participants in group 2 and group 3 were compared separately with participants in group 1 (the control group).”

VERSION 2 – REVIEW

REVIEWER	Kevin C Maki Midwest Biomedical Research USA I have conducted clinical trials and published in the area of cranberry and UTI sponsored by Ocean Spray Cranberries.
REVIEW RETURNED	10-Feb-2021

GENERAL COMMENTS	Thank you for addressing my comments and requests for clarification. I have no further comments.
--

REVIEWER	Holly Fisher Newcastle University, UK
REVIEW RETURNED	01-Feb-2021

GENERAL COMMENTS	I have no further comments on this manuscript. Congratulations to the authors on a very well conducted and written up feasibility trial.
--